# L-Poly(lactic acid) Production by Microwave Irradiation of Lactic Acid Obtained from Lignocellulosic Wastes

**DOI:** 10.3390/ijms24129817

**Published:** 2023-06-06

**Authors:** Lacrimioara Senila, Oana Cadar, Eniko Kovacs, Emese Gal, Monica Dan, Zamfira Stupar, Dorina Simedru, Marin Senila, Cecilia Roman

**Affiliations:** 1Research Institute for Analytical Instrumentation Subsidiary, National Institute for Research and Development of Optoelectronics Bucharest INOE 2000, 67 Donath Street, 400293 Cluj-Napoca, Romania; oana.cadar@icia.ro (O.C.); eniko.kovacs@icia.ro (E.K.); zamfira.stupar@icia.ro (Z.S.); dorina.simedru@icia.ro (D.S.); marin.senila@icia.ro (M.S.); cecilia.roman@icia.ro (C.R.); 2Faculty of Horticulture, University of Agricultural Sciences and Veterinary Medicine, 3-5 Manastur Street, 400372 Cluj-Napoca, Romania; 3Faculty of Chemistry and Chemical Engineering, Babes-Bolyai University, 11 Arany Janos Street, 400028 Cluj-Napoca, Romania; emese.gal@ubbcluj.ro; 4National Institute for Research and Development of Isotopic and Molecular Technologies, 67-103 Donath Street, 400293 Cluj-Napoca, Romania; monica.dan@itim-cj.ro

**Keywords:** L-polylactic acid, renewable biomass, simultaneous saccharification and fermentation, lactic acid

## Abstract

L-polylactic acid (PLA), a semi–crystalline aliphatic polyester, is one of the most manufactured biodegradable plastics worldwide. The objective of the study was to obtain L-polylactic acid (PLA) from lignocellulosic plum biomass. Initially, the biomass was processed via pressurized hot water pretreatment at a temperature of 180 °C for 30 min at 10 MPa for carbohydrate separation. Cellulase and the beta-glucosidase enzymes were then added, and the mixture was fermented with *Lacticaseibacillus rhamnosus* ATCC 7469. The resulting lactic acid was concentrated and purified after ammonium sulphate and n-butanol extraction. The productivity of L-lactic acid was 2.04 ± 0.18 g/L/h. Then, the PLA was synthesized in two stages. Firstly, lactic acid was subjected to azeotropic dehydration at 140 °C for 24 h in the presence of xylene, using SnCl_2_ (0.4 wt.%) as a catalyst, resulting in lactide (CPLA). Secondly, microwave-assisted polymerization was carried out at 140 °C for 30 min with 0.4 wt.% SnCl_2_. The resulting powder was purified with methanol to produce PLA with 92.1% yield. The obtained PLA was confirmed using electrospray ionization mass spectrometry, nuclear magnetic resonance, thermogravimetric analysis, Fourier transform infrared spectroscopy, scanning electron microscopy, and X-ray diffraction. Overall, the resulting PLA can successfully replace the traditional synthetic polymers used in the packaging industry.

## 1. Introduction

The production of bioplastics from renewable materials has captured considerable research interests as an opportunity to mitigate greenhouse gas emissions, which have resulted mainly from the combustion of fossil fuels and plastics. Renewable biomass can be used to produce fuels and chemicals as a substitute for oil resources. To produce bioplastics from lignocellulosic feedstocks, the main components of polysaccharides (cellulose and hemicellulose) must be extracted and subjected to pretreatment and hydrolysis [1]. Lignocelluloses are mainly composed of cellulose (25–55%), hemicellulose (11–50%), and lignin (10–40%). Fermentable sugars are utilized as a substrate in the fermentation process for the production of biopolymers, such as poly(lactic acid) (PLA) [2]. The main components of cellulosic biomass (cellulose and hemicellulose) can be converted into PLA, while cellulose and lignin can be transformed into polyhydroxyalkanoate (PHA). Cellulose is a homopolymer of β-D-glucose, with structural units connected by *β*–1–4 glycoside bonds and the chemical formula of (C_6_H_10_O_5_)_n_, where n ranges between 700 and 800 and 2500 and 3000. Hemicelluloses are polysaccharides of pentoses (xylose and arabinose), hexoses (mannose, glucose, and galactose), and uronic acids. Lignin is a complex polymer comprising three types of precursor alcohols: p-coumaric alcohol, coniferous alcohol, and p-synaptic alcohol. Monomeric sugar may be fermented using fermentative microorganisms to produce bioplastics [1,3]. 

The conversion of biomass into bioplastics involves the following steps: biomass pretreatment, involving the separation of biomass into components; hydrolysis of cellulose into monomeric sugars; fermentation of lactic acid; and polymerization [4]. In order to minimize the processing steps, hydrolysis and fermentation can be used simultaneously for the direct conversion of cellulosic waste into lactic acid via simultaneous saccharification and fermentation (SSF) [5,6,7]. The pretreatment methods include physical, chemical, biological, and thermochemical approaches. Moreover, in order to reduce reactive consumption and for environmental reasons, steam-explosion, liquid hot water, microwave processes, and CO_2_ explosion pretreatment can also be applied [8].

Poly(lactic acid) (PLA), a semi-crystalline aliphatic polyester, is the second most manufactured biodegradable plastic, with a global production share of 13.9% [9]. PLA is a competitive polymer among the conventional plastics, such as low-density polyethylene (LDPE) and polyethylene (PP), due to its superior mechanical strength, durability, and clarity [10]. Biodegradable PLA is also used to produce degradable packaging materials for food, mulch films, household bags, rigid containers, and other bioplastics. Due to its biodegradable, biocompatible, and bioabsorbable properties, PLA has applications in medicine. Products such as porous scaffolds, orthopedics, medicine carriers, nanofibers, foams, biocomposites, and personal protective equipment and medical supplies are manufactured from PLA [11,12].

Lactic acid bacteria (LAB) are a heterogeneous bacterial group frequently present in milk and fermented dairy, vegetable products, water, soil, and the gastrointestinal tract of humans and animals. This group of bacteria is well known for its ability to ferment carbohydrates to produce lactic acid, and is widely used in the fermented food industry [13]. LAB are gram-positive bacteria that have a cocci or rods shape, are nonmotile, anaerobic, catalase-negative, and typically do not form spores [14]. Although there are about 60 LAB genera, the most commonly encountered in food fermentation are *Lactobacillus*, *Lactococcus*, *Leuconostoc*, *Pediococcus*, *Streptococcus*, *Enterococcus*, *Weissella*, etc. [15].

Due to the large phenotypic, ecological, and genotypic diversity, the genus *Lactobacillus* was recently reclassified into 25 genera (including the previously considered *Lactobacillus* species) and 23 new genera [16]. Among the genus *Lactobacillus*, species *Lactobacillus delbrueckii*, *Lactobacillus bulgaricus*, *Lactobacillus leichmanii*, *Lactobacillus plantarum* (formerly *Lactobacillus plantarum*), and *Lacticaseibacillus rhamnosus* (formerly *Lactobacillus rhamnosus*), with other homolactic fermentation strains, are employed to produce lactic acid [17].

The most commonly used methods for purifying lactic acid are precipitation with calcium hydroxide, solvent extraction (aliphatic amine, chloroform, hexane, decanol, diethyl ether), salting-out extraction, reactive extraction, emulsion liquid membrane, membrane separation, electrodialysis, molecular distillation, and esterification [2]. The demand for pure lactic acid (L or D isomer) in various sectors is promoted by the microbial fermentation pathway in the chemical route. PLA is commonly produced by polycondensation and ring-opening polymerization to generate a high molecular weight. Generally, supplementary procedures are required to obtain PLA with a high molecular weight, such as solid state polycondensation, chain extension, etc. Many works have reported on the classical methods for the production of PLA. In this regard, Hu et al. produced PLA by ring-opening polymerization using aqueous ZnO nanoparticles as catalysts, and the lactide precursor was obtained with a 91–92% yield after a reaction time of 8 h [18]. Ahmad et al. reported the polymerization of lactic acid produced from food waste using SnCl_2_ and FePO_4_ nanocrystals as catalysts for lactide polymerization [19]. The reaction took place at 220 °C for 10–12 h. In PLA production, *Diutina rugosa* (formerly *Candida rugosa*) lipase at a concentration of 2% was also used as a biocatalyst for L-lactide polymerization at 90 °C [20,21]. Metal complexes, alcohols, and other compounds are used as initiators of ring opening polymerization. In general, PLA is synthesized by polycondensation and the ring-opening polymerization of lactic acid, as well as by direct polycondensation at high temperatures, with or without the use of catalysts. The catalysts used are metal complexes (Al, Mg, Zn, Ca, Sn, Fe, Y, Sm, Lu, Ti, and Zn). Some studies also reported the production of PLA from various raw materials by polycondensation of lactic acid through microwave irradiation [22,23]. High molecular weight L-poly(lactic acid) (PLA) was synthetized by the ring-opening polymerization of L-lactide in the presence of 1,12-dodecadiol and ditrimethylolpropane as initiators [24].

The current study aims to investigate (a) the pretreatment of lignocellulosic biomass in order to separate the cellulosic components; (b) the simultaneous saccharification and fermentation of cellulose separated from lignocellulosic waste to produce lactic acid; (c) poly(lactic acid) production from L-lactic acid by microwave irradiation; and (d) the structural characterization of the polymer using electrospray ionization mass spectrometry (ESI–MS), proton nuclear magnetic resonance (^1^H-RMN), X-ray diffraction (XRD), thermogravimetric analysis (TGA), Fourier transform infrared (FTIR) spectroscopy, and scanning electron microscopy (SEM). Our previous publication described a method for cellulose separation from apple orchard wastes by supercritical carbon dioxide [25]. In this regard, the present study continues the investigation in order to obtain lactic acid from cellulose separated from plum orchard waste, and, finally, the polycondensation of lactic acid to PLA using a more environmentally friendly method.

## 2. Results and Discussion

### 2.1. Pretreatment of Raw Material

The raw material composition before and after pretreatment is presented in Table 1.

The plum orchard biomass contains 38.4 ± 1.2% cellulose, 26.8 ± 1.0% hemicelluloses, and 28.6 ± 0.98% lignin. The pretreatment method was used for the separation of cellulose constituents. The solid fraction resulting after the pretreatment contained cellulose, lignin, and small quantities of hemicelluloses (8.1 ± 0.04%). Hemicelluloses were recovered in the liquid fraction as mixtures of sugars (xylose, glucose, mannose, galactose, and arabinose) and secondary byproducts (5-hydroxymethyl furfural and furfural) due to their hydrophilic character. Biomass has a hydrophilic character due to the large number of hydroxyl groups from cellulose and hemicellulose. After pretreatment, the hydrophobic character was obtained as a result of the elimination of hemicelluloses in the water extract. This property is needed for the subsequent enzymatic hydrolysis process. The solid yield obtained after the pretreatment method was 62.2%. According to the literature, eliminating inhibitory compounds prior to enzymatic hydrolysis and using different lignin blockers could improve the performance of enzymatic hydrolysis [25]. The high cellulose content (48.5 ± 1.6 g/100 g pretreated biomass) obtained from the pretreated biomass is recommended as a substrate for microbial fermentation. A delignification method was applied in order to separate only the cellulose. The solid yield of delignification was 42.5 ± 2.4%. The delignified solid contains 98% cellulose. The pretreated and delignified solids were used for the SSF process.

### 2.2. SSF Process

The pretreated and delignified biomass from the plum orchard was used in the SSF process for lactic acid production. Cellulose can be degraded by a group of enzymes that contain the cellulase complex: endoglucanase, exoglucanase, and β-glucosidase [26]. In this sense, *Trichoderma reesei* and β-glucosidase were used in this research, and the recommended temperature for hydrolysis was 50 °C at a pH of 5–5.5. The strain *Lacticaseibacillus rhamnosus* ATCC 7469, used for fermentation, can produce only L(+)-lactic acid at a recommended temperature of 37 °C at a pH of 5–6. In the SSF process, enzymatic hydrolysis and fermentation were carried out simultaneously in the same bioreactor. The cellulose can be hydrolyzed into glucose which can then be fermented to lactic acid. The SSF process was initiated by simultaneously adding enzyme mixtures and microbial inoculum. The lactic acid production during the SSF of the delignified biomass is presented in Figure 1. In order to avoid the formation of enzyme inhibitors, the enzymatic hydrolysis and fermentation occur simultaneously. 

According to Figure 1, the highest lactic acid was produced (94.0 ± 3.1 g/L) at 37 °C and at a pH of 5.5, corresponding to 0.86 ± 0.04 g lactic acid/g of glucose after 48 h of fermentation. The experiments were carried out at 50 °C to favor the cellulase enzyme activities, demonstrating that lactic acid is produced in a low concentration (below 5.1 ± 0.1 g/L) in both cases of the tested pH values. Karnaouri et al. reached the same conclusion regarding D-lactic production by the *Lactobacillus delbrueckii* species [17]. The largest amount of L-LA was produced in the first 48 h of the fermentation process. At the end of the process, the pH of the broth was 4.75. The lactic acid recovery was 90.2%. After 48 h, the residual reducing sugar was 0.73 ± 0.02%. 

According to a study conducted by Chen et al., a high cellulase loading in the SSF process substantially improved the enzymatic hydrolysis [27]. In total, 25 FPU/g was used in this study to enhance the enzymatic hydrolysis of celluloses and further fermentation to L-LA acid. Bahry et al. reported that lactic acid production from carob pod waste using *L. rhamnosus* encapsulated in alginate beads generated 22 g/L, a yield of 76.9%, and a productivity of 1.22 g/L/h from 65 g/L total sugars [28]. Besides lactic acid, impurities of acetic, citric, formic, propionic, and succinic acid were also identified. 

In the SSF of the pretreated biomass, 53.0 ± 1.8 g/L of potential glucose was introduced in a bioreactor. The presence of a small hemicellulose quantity in the pretreated biomass can be hydrolyzed first and act as an inhibitor of enzymatic hydrolysis. After 24 h of fermentation, 45.95 ± 2.1 g/L of lactic acid was produced at an efficiency of 73.3% (taking into account hemicelluloses). After 44 h of fermentation, glucose was transformed into lactic acid. A pH of 5.5 and a temperature of 50 °C were adequate for producing high lactic acid. By comparing the delignified and pretreated biomass substrates used in the SSF process, it can be inferred that the pretreated substrate can produce a substantial quantity of lactic acid, without requiring a prior delignification method. The SSF process for lactic acid has the benefit of producing lactic acid from pretreated biomass compared to the SSF process for ethanol production, where a delignification method has substantially improved the ethanol yield [24].

The mass balance for L-LA production from lignocellulosic biomass is presented in Figure 2. From 100 ± 3.2 g of lignocellulosic biomass, 65 ± 2.6 g of the solid phase was separated after pressurized hot water pretreatment. The SSF process was finalized by producing 29.2 ± 1.1 g of L-LA. The resulting L-LA was further used to produce L-poly(lactic acid) (PLA).

In Table 2, the lactic acid concentrations obtained in the current study are compared with those reported in the literature.

### 2.3. Microwave Assisted Polymerization of Lactic Acid to PLA

The polymerization of lactic acid into PLA takes place in two steps: (a) azeotropic dehydration for 24 h and (b) microwave-assisted polymerization at 140 °C and for 30 min to obtain a high molecular mass of PLA. Water removal was carried out by azeotropic distillation using oxylene as solvent (i.e., initiator of the reaction) and SnCl_2_ as catalyst (0.4 wt.%). Considering that obtaining high molecular-weight PLA by direct polycondensation is difficult [32], the elimination of water was performed using a Dean–Stark trap and molecular sieves before microwave irradiation was applied. The obtained PLA (PLA 1) was purified by precipitation in cold methanol with a yield of 92.1%.

### 2.4. Structural Characterization of PLA

#### 2.4.1. ESI (+)–HRMS Spectra

The chemical structure of linear PLA is presented in Figure 3. Figure 4 shows an ESI (+)–HRMS spectrum of sample PLA 2. The most abundant form obtained in the polymerization reaction is PLA, with a polymerization degree of n = 4–29 [M_n_+H_2_O+Na]^+^. In the mass spectrum, K^+^ adducts were also detected. CPLA [M_n_+Na]^+^ adducts were also found in low quantities as a byproduct. The measured m/z for [M_12_+H_2_O+Na]^+^ is 905.2519, and the calculated *m*/*z* is 905.2533, yielding a difference in ppm of 1.55, which confirms the structure of the proposed poly(lactic acid). M12 indicates the polymer with a polymerization degree of n = 12. In the case of PLA standard, in the ESI–MS spectrum (Figure 5), a 615 *m*/*z* fragment appears, which can be attributed to the [M_16_+CH_3_OH+2Na]^2+^ adduct [33]. The presence of alkali metals originates from the ambient contaminants.

In the PLA 1 sample, ESI–HRMS spectra fragments which occur in the case of linear polylactic acid PLA 2 can be observed. In addition, fragments such as *m*/*z* 615 are present in the mass spectra. These can be attributed to the adduct with solvent and sodium [M_16_+CH_3_OH+2Na]^2+^. The fragments corresponding to the CPLA molecule are present in the MS spectrum, in this case with higher intensity compared to the PLA 2 sample. The ESI–MS spectrum of PLA 1 is presented in Figure 6.

#### 2.4.2. Proton Nuclear Magnetic Resonance (^1^H-RMN)

The PLA obtained was confirmed by ^1^H-RMN spectroscopy. According to the structure presented in Figure 1, PLA 1 presents two groups, -OH and -COOH, at the end of the structure. The ^1^H-RMN spectra of the PLA, before and after purification, and CPLA are presented in Figure 7a–c. The resonance of the signal present at 1.56–1.58 ppm (d, 3H, and CH_3_) was attributed to the methyl protons. The signal present at 5.12–5.18 ppm was attributed to the methine protons (-CH) (q, 1H, and CH). The PLA was formed by CPLA polymerization, as confirmed by the ^1^H-RMN of CPLA (Figure 7c). The spectrum presents two signals: 5.13–5.22 ppm, attributed to -CH, and 1.53–1.58 ppm, attributed to the CH_3_ groups from the lactide structure. The confirmed structure of PLA is in accordance with the study of Suganuma et al., which confirms the NMR analysis of poly(lactic acid) via a statistical model [34].

The ^1^H-RMN spectra were used to estimate the molecular weight of the obtained PLA. The molecular weight of PLA 1 was expressed as an average because the polymer contains a mixture of atoms of different lengths. The average molecular weight of the PLA (M_n_) was estimated according to Phuphuak et al. [35] and Viamonte-Aristizábal et al. [21]. The terminal units contain a hydroxyl group and carboxylic acid (Figure 8). The ^1^H-RMN spectra of the PLA 1 and PLA 2 samples are presented in Figure 9.

Signals located at δ_H_ = 5.19 and 1.60 ppm were assigned to the methine (CH, c) and methyl (CH_3_, a) groups in the polymer; meanwhile, the signal located at δ_H_ = 4.37 ppm was attributed to the methine end-group (CH, b).
(1)$$M_{n}=\left(M_{w\;reapiting\;unit}\;\cdot \;n\right)+\left(M_{w\;end\;unit}\right)$$
(2)$$n=\frac{c}{b}$$
where $M_{w\;reapiting\;unit}\;\mathrm{is}\;\mathrm{the}\;\mathrm{mass}\;\mathrm{of}\;\mathrm{eparative}\;\mathrm{units},\;\;M_{w\;end\;unit}\;$ is the mass of terminal units, *c* is the intensity of the signal in the proton spectrum of the methine end-group, and *b* is the intensity of the signal in the proton spectrum of methine from the repeating unit.

The *n* of the PLA structure was calculated from the ^1^H-RMN abundance and by dividing the intensity of the signal in the proton spectrum of the methine end-group and the methine proton from the repeating unit peaks. The intensity of the methine proton signal located at 4.37 ppm was significantly lower than the internal methine. The n calculated for PLA 1 was 22.12 and was approximated to 22, and the calculated *n* for PLA 2 was 14.10 and was approximated to 14. The *M_n_* of the PLA calculations was performed as follows:$$M_{PLA\;1}=\left(72\;\cdot \;22\right)+\left(72+17+1\right)=1674\;g/\mathrm{mol}$$
$$M_{PLA\;2}=\left(72\;\cdot \;14\right)+\left(72+17+1\right)=1098\;g/\mathrm{mol}$$

#### 2.4.3. FTIR Spectrum

The FTIR spectrum for PLA 1 obtained after purification is presented in Figure 10. The chemical structure of PLA was confirmed by the FTIR scan between 4000 and 600 cm^−1^. The structure presents the following bands: 1749 cm^−1^ attributed to the carbonyl stretching (C=O), 1455 cm^−1^ is the CH_3_ asymmetrical scissoring band, 1188 cm^−1^ attributed to the C-O and C-O-C vibration, 1045 cm^−1^ is the band of C-CH_3_, and 870 cm^−1^ is the band of C-COO. Two -C-H bonds were obtained at 2999 cm^−1^ (asymmetric) and 2948 cm^−1^ (symmetric). The -OH bond attributed to carboxylic acid occurred at 3500 cm^−1^. The -C=O and OH bonds were identified at 1216 cm^−1^ and 1082 cm^−1^. The distinct peak at 1361 cm^−1^ indicated the semi-crystalline structure of the obtained polymer. The same FTIR spectrum was obtained by Rahmayetty et al. for the characterization of polylactic acid production by the ring-opening polymerization of L-lactide by the use of *Diutina rugosa* [20]. The FTIR spectrum confirmed the vibration of the obtained PLA. The results are in accordance with the FTIR spectra of the PLA obtained by Tacsi et al., who reported the production and characterization of polylactic acid microparticles using electrospray with porous structures [10]. 

#### 2.4.4. TGA Analysis

Thermogravimetric analysis (TGA) provides information about the weight loss step. Figure 11 presents the TGA/DTG curves of the PLA samples. The thermal degradation of PLA, before and after purification, and lactide takes place in one stage. PLA purification has a significant effect on the thermal degradation by enhancing thermal stability. Based on the TGA curve, no significant loss of biomass was obtained for PLA 1, while a significant loss of biomass was obtained for the CPLA product. The thermal degradation of PLA begins around 300 °C and ends around 400 °C. The maximum degradation temperature of PLA before purification was 316.45 °C, whereas the temperature of PLA after purification increased to 361.62 °C. The degradation temperature of lactide was 253.2 °C. These higher degradation temperatures for PLA, before and after purification, could be attributed to an increase in PLA molecular weight and to a good thermostability compared with the TGA/DTG of CPLA, which show a low thermostability. The PLA compound has only one degradation stage and it corresponds to the cleavage of the bonds on the polymer with the degradation of lactide, oligomers, acetaldehyde, and carbon monoxide. Some loss of water prior to 200 °C was observed for PLA 2 (Figure 11). The high stability of PLA 1 results from the presence of hydroxide groups [36]. The TGA for PLA demonstrated that it can be used to replace the synthetic polymer used in packaging.

#### 2.4.5. XRD Analysis

The similar XRD patterns of purified and unpurified PLA are presented in Figure 12. The characteristic diffraction peaks in 2θ angles positioned at 2θ 12.42, 14.76, 16.45, 19.02, 22.04, 23.91, 29.04, and 31.20°, matching the reflection planes (103), (104), (200), (203), (211), (213), (310), and (217), confirm the presence of a pure, crystallized PLA phase of orthorhombic structure (PDF 00-064-1624). Generally, polymers can comprise crystalline regions randomly mixed with amorphous regions. Moreover, high intensity peaks indicate a polymer with a crystalline structure, while an amorphous structure results in ramps and wider bands [20]. The degree of crystallinity (75% for PLA 1 and 70% for PLA 2), calculated as the ratio between the crystalline regions over the sum of crystalline and amorphous area [37], was considerably higher than in the PLA obtained by the ring-opening polymerization of L-lactide using *Diutina rugosa* lipase [20].

#### 2.4.6. Scanning Electron Microscopy (SEM)

The SEM analysis of the PLA products obtained is presented in Figure 13. Energy dispersive X-ray spectroscopy (EDX) was used to determine the elemental composition of PLA 1 (the final product) and revealed a content of 65.01% C and 34.99% O. The SEM morphology of PLA 1 shows fine particle sizes of a smooth surface with uniform pores. The elemental compositions of CPLA show 58.96% C and 41.04% O. The morphology of CPLA (Figure 13c) shows an instable structure with cracks and large voids. There is a notable difference among the three structures. The surface area of the fiber will grow as the roughness increases [38]. 

## 3. Materials and Methods

### 3.1. Chemicals and Reagents

All the chemicals used were analytical reagent grade. Acetic acid (C_2_H_4_O_2_), dichloromethane (CH_2_Cl_2_), methanol (CH_4_O), D(+)-glucose (C_6_H_12_O_6_), stannous chloride (SnCl_2_), o-xylene 98% (C_8_H_10_), calcium carbonate (CaCO_3_), *S. cerevisiae* YSC2 (YSC_2_), enzymatic digest of casein, meat extract, sodium acetate (C_2_H_3_NaO_2_), diammonium citrate (C_6_H_14_N_2_O_7_), dipotassium phosphate (K_2_HPO_4_), magnesium sulphate (MgSO_4_), iron (III) chloride hexahydrate (FeCl_3_·6H_2_O), 1-butanol (C_4_H_10_O), ammonium sulphate ((NH_4_)_2_SO_4_), molecular sieve 3 Å (K_n_Na_12-n_[(AlO_2_)_12_(SiO_2_)_12_]·xH_2_O), 3.5-dinitrisalycilic acid (DNS), and sodium hydroxide (NaOH) were purchased from Merck (Darmstadt, Germany). Sodium chlorite (80%) (NaClO_2_) was purchased from Alfa Aesar GmbH & Co. (Karlsruhe, Germany). Enzymes cellulase from *Trichoderma reesei* ATCC 26921 and β-glucosidase from almonds were purchased from Sigma-Aldrich (St. Louis, MO, USA). *L. rhamnosus* ATCC 7469 was purchased from Microbiologics (Cooper Avenue North, St. Cloud, MN, USA). All solutions were prepared by using ultrapure water (18.2 MΩcm^−1^ at 20 °C) obtained from a Direct-Q3 UV Water Purification System (Millipore, Molsheim, France).

### 3.2. Sample Description

The plum orchard biomass was purchased from the Research Station of the University of Agricultural Sciences “Ion Ionescu de la Brad” in Iasi, Romania. The biomass contained plum tree branches and trunks obtained after orchard pruning. The samples were dried at 105 °C and shredded to a diameter of 0.2 mm.

### 3.3. Pressurized Hot Water Pretreatment of Raw Biomass

Pressurized hot water pretreatment was carried out according to our previous publication method with modifications [24]. The raw material (30 g) and 270 mL of water were introduced in a Parr reactor (Parr Instruments, Moline, IL, USA), equipped with a temperature controller and a 1 L reaction vessel, and heated at a temperature of 180 °C for 30 min at 10 MPa. The solid fraction was separated by filtration and analyzed for cellulose, lignin, and hemicellulose content. 

### 3.4. Delignification of Pretreated Biomass

The solid fraction resulting from the pretreatment method was subjected to delignification with sodium chlorite. About 5 g of the pretreated biomass was mixed with 250 mL of acetic acid 10% and 5 g of NaClO_2_. The mixture was heated at 80 °C for 4 h. The solid fraction was separated by filtration and analyzed in regard to the cellulose and residual lignin content.

### 3.5. Simultaneous Saccharification and Fermentation Process SSF to Lactic Acid

The solid fractions that resulted after pretreatment and delignification were introduced in a 1.7 L bioreactor (Lambda Minifor, Lambda Laboratory Instruments, Brno, Czech Republic) equipped with a sensor for dissolved oxygen, pH, and temperature. The solid loading was 10%, and the nutrient (like MRS without glucose) was sterilized at 121 °C for 15 min. CaCO_3_ (4%) was added before adjusting the pH. The cellulase from *Trichoderma reesei* ATCC 26921 and β-glucosidase from almonds were used for the enzymatic hydrolysis of cellulose into glucose. In all the experiments, a 25 FPU/g substrate of enzymes from *Trichoderma reesei* and 20 U/g of β-glucosidase were used. The inoculum of *L. rhamnosus* was 10% (*v*/*v*). The experiments were conducted at two temperatures: 37 °C and 44 °C for 72 h. Additionally, two pH values were tested: 5.5 and 6.5. During each experiment, 1 mL of sample was taken and boiled for 5 min until the enzymes’ deactivation, centrifuged at 5000 rpm for 10 min in order to remove the precipitate, and then used for lactic acid and residual sugar analysis. 

#### Preparation of Stock Culture and Cultivation Conditions

The strain used for the fermentation of cellulose was *L. rhamnosus* ATCC 7469. A pre-inoculum of *L. rhamnosus* ATCC 7469 was prepared in sterilized (at 121 °C for 15 min.) broth medium containing 20 g/L glucose, 5 g/L yeast extract, 10 g/L enzymatic digest of casein, 10 g/L meat extract, 5 g/L sodium acetate, 2 g/L diammonium citrate, 2 g/L K_2_HPO_4_, and 0.2 g/L MgSO_4_. The inoculated medium was incubated at 37 °C, in aerobic conditions for 48 h, and stored at 4 °C as a stock culture. Every week, a fresh working culture was prepared from the stock culture in order to maintain the viability of the strain. For the fermentation step, a fresh *L. rhamnosus* inoculum was prepared from the working culture in 5 mL NaCl at an optical density (OD) of OD600 nm = 4.

### 3.6. Purification of Lactic Acid

The fermentation broth was extracted with a mixture of ammonium sulfate and n-butanol according to the method of Kumar et al. [39]. The fermentation broth was treated with 10 g of ammonium sulfate and then 50 mL of n-butanol was added to the mixture and stirred. The organic phase was separated and evaporated to dryness.

### 3.7. Synthesis of PLA

About 20 g of L-lactic acid obtained by fermentation was introduced in a 250 mL reaction vessel equipped with a Dean–Stark trap. SnCl_2_ was used as a catalyst (0.4 wt.%) and 30 mL of xylene was added to the mixtures. The mixtures were refluxed at 140 °C for 24 h under controlled agitation in order to remove the water. The molecular sieves were used as drying agents. After the complete elimination of water, the mixtures were completely distilled. The obtained lactide (CPLA) was purified by recrystallization in ethyl acetate, dried, and heated using microwave irradiation in a microwave reactor (Synthos 3000, Anton Paar, Australia) at 140 °C for 30 min at a maximum pressure of 60 bar with 0.4 wt.% SnCl_2_. The resulting mixture was poured into cold methanol, and the white precipitate (PLA 2) was filtered and dried under reduced pressure. The recrystallization of PLA 2 in methanol led to purified PLA 1 [18].

### 3.8. Chemical Characterization

#### 3.8.1. Chemical Characterization of Raw and Pretreated Biomass

The contents of cellulose, hemicelluloses, and lignin were determined according to Teramoto et al. [40].

#### 3.8.2. Determination of Reducing Sugars

The concentration of reducing sugar was determined by using a Lambda 25 ultraviolet visible spectrometer (Perkin Elmer), according to the Miller method [41]. The DNS reagent was prepared by mixing 40 mL of water with 1 g of 3.5-dinitrisalycilic acid and 2.075 mL of NaOH 50%. The obtained solution was mixed with 30 g of potassium and sodium tartrate, and the final volume of the solution was 100 mL with water. A quantity of 1 mL of hydrolysate solution was mixed with 3 mL of DNS reagent, and the solution was boiled for 5 min. The solution absorbance was measured at 540 nm. The reduced sugars were calculated using Equation (3):Reduced sugars (%) = W_S_/W_1_ (1 × 1000) × 100(3)
where W_S_ is the quantity of sugars determined by reaction with DNS (mg) and W_1_ is the initial sample (g). WS is calculated with Equation (4):WS = C_DNS_ × V(4)
where C_DNS_ is the concentration of sugars (mg/mL), V is the total volume of hydrolysates (mL), and 0.9 is the conversion factor of the cellulose transformation into glucose.

#### 3.8.3. Determination of Lactic Acid Concentration

The lactic acid produced by the SSF process was determined at a 390 nm wavelength using a Lambda 25 spectrophotometer (Perkin Elmer, Beaconsfield, UK) with 1 cm glass cells, according to Borshchevskaya et al. [42]. Lactic acid (90%) was used as a standard for the calibration curve at a concentration ranging between 0.5 and 10 g/L, by diluting the stock solution with ultrapure water. A volume of 50 μL of cultural liquid was treated with 2 mL of 0.2% FeCl_3_·6H_2_O solution, and then the absorbance at 390 nm was measured.

### 3.9. Structural Characterization of the Obtained PLA

#### 3.9.1. Electrospray Ionization Mass Spectrometry (ESI–MS), Proton Nuclear Magnetic Resonance (^1^H-RMN)

NMR spectra were recorded at room temperature on a Bruker Advance instrument (^1^H: 400 MHz) using a CDCl_3_ solvent. The NMR software MestReNova was used for the data analysis and processing. The sample (0.3 g) was dried in a vacuum oven at 40 °C and then dissolved in deuterated CDCl_3_ (1 mL) before characterization. HRMS spectra were recorded on a Thermo Scientific LTQ Orbitrap XL. The samples were dissolved in anhydrous methanol before analysis.

#### 3.9.2. TGA/DTG Analysis

The thermal decomposition and derivative thermogravimetric (DTG) analysis of PLA and lactide were determined using a TA Instruments SDT O 600 (TA Instruments, New Castle, DE, USA) at temperatures ranging from 30 to 1000 °C at 10 °C per minute under air. A mass of 7.78 ± 0.3 mg of the dried samples was used for testing, and the experiment was conducted under a nitrogen atmosphere. The experiments were repeatable with a standard deviation in the peak temperature values.

#### 3.9.3. X-ray Diffraction (XRD)

X-ray diffraction (XRD) patterns were recorded on a D8 Advance diffractometer (Bruker AXS, Karlsruhe, Germany) with a CuKα anode (λ = 1.5406 Å), operating at 40 kV and 35 mA, at room temperature.

#### 3.9.4. FTIR Spectroscopy

The FTIR spectrum of PLA was recorded using a Bruker Vector 22 FT-IR spectrometer (Bruker, Billerica, MA, USA) in the range of 4000–600 cm^−1^ on 1% KBr pellets in order to identify different functional groups and monitor changes occurring at the functional group level during different conversion processes.

#### 3.9.5. SEM Analysis

The morphology of PLA before and after purification was examined using a scanning electron microscope (SEM VEGAS 3 SBU, Tescan, Brno-Kohoutovice, Czech Republic) with an EDX detector. The samples were deposited on double-sided conductive carbon tape on aluminum stubs and analyzed.

## 4. Conclusions

In this study, plum orchard biomass was used for the first time as a raw material for poly(lactic acid) production by the fermentative pathway of carbohydrates obtained by supercritical CO_2_ extraction. The solid fraction that resulted after the pretreatment was used for lactic acid fermentation in the SSF process, utilizing *Lactocaseibacillus rhamnosus* ATCC 7469. The experimental data proved that the microwave irradiation of lactic acid substantially improved the molecular weight of PLA. The study confirmed that plum orchard biomass can be converted into L-poly(lactic acid) via the following steps: pretreatment, simultaneous saccharification and fermentation processes, polycondensation and polymerization under microwave irradiation, and, finally, purification. Thermogravimetric analysis showed a good thermostability of the purified PLA. The structure of the obtained product was confirmed by electrospray ionization mass spectrometry and proton nuclear magnetic resonance. The ESI–MS spectra of PLA 1 and CPLA included Na^+^ and K^+^ adduct ions. The integration of the proposed process into the ecological and sustainable production of valuable products would make a significant contribution to the development of a cleaner chemical industry.

## Figures and Tables

**Figure 1 ijms-24-09817-f001:**
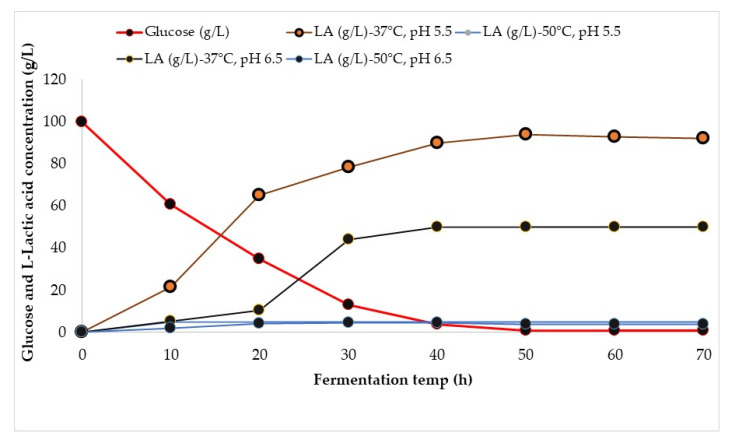
Lactic acid formed during the SSF process of the delignified biomass.

**Figure 2 ijms-24-09817-f002:**
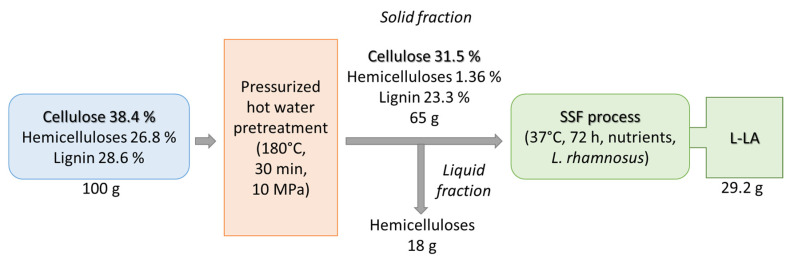
Mass balance for the production of lactic acid from lignocellulosic biomass.

**Figure 3 ijms-24-09817-f003:**
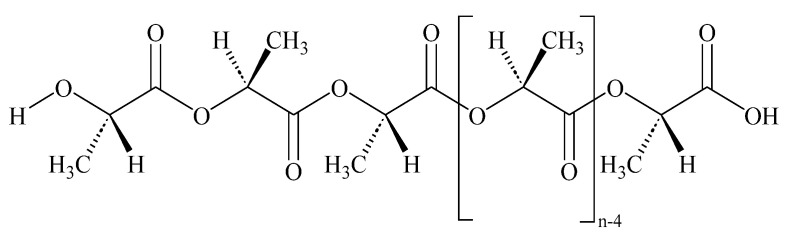
Chemical structure of linear PLA.

**Figure 4 ijms-24-09817-f004:**
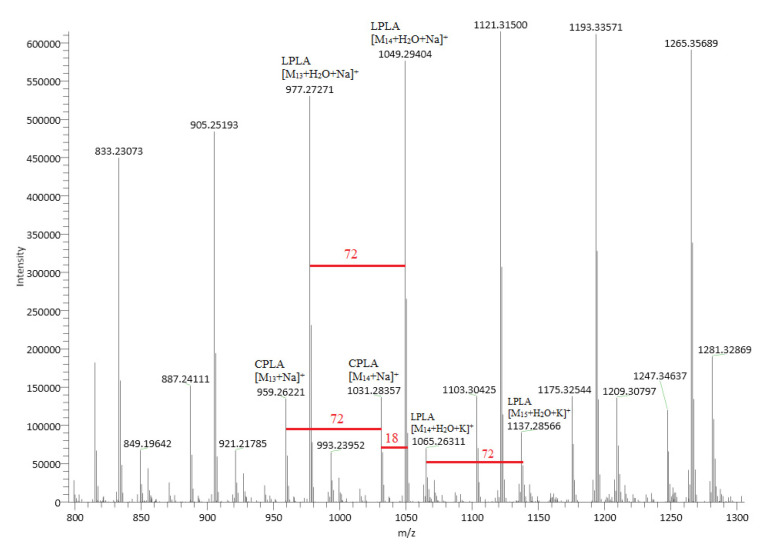
ESI (+)–HRMS spectrum of sample PLA 2, with the specific adducts. The small quantity of CPLA is a byproduct. M13 indicates the polymer with a polymerization degree of n = 13.

**Figure 5 ijms-24-09817-f005:**
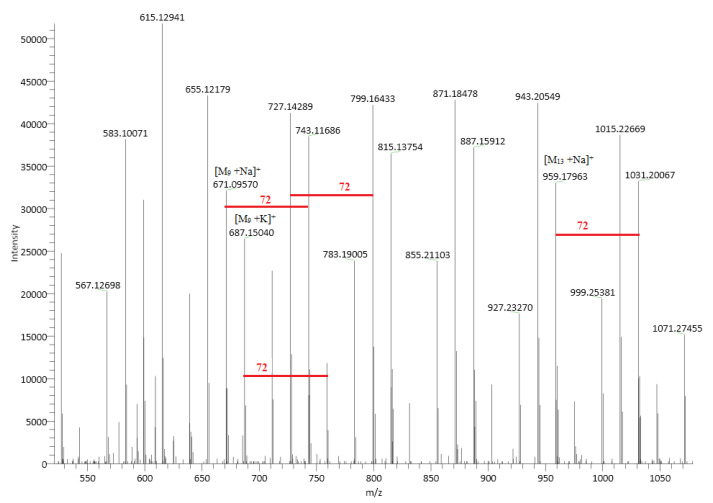
ESI–MS spectrum of a PLA standard sample.

**Figure 6 ijms-24-09817-f006:**
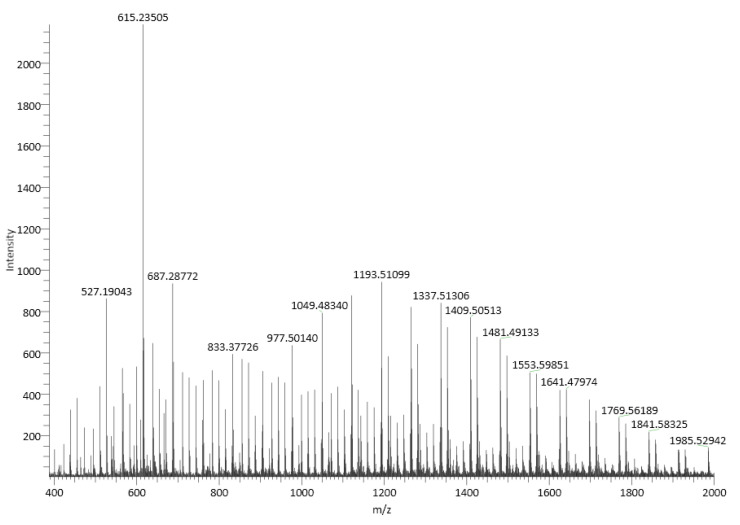
ESI–MS spectrum of the PLA 1 sample.

**Figure 7 ijms-24-09817-f007:**
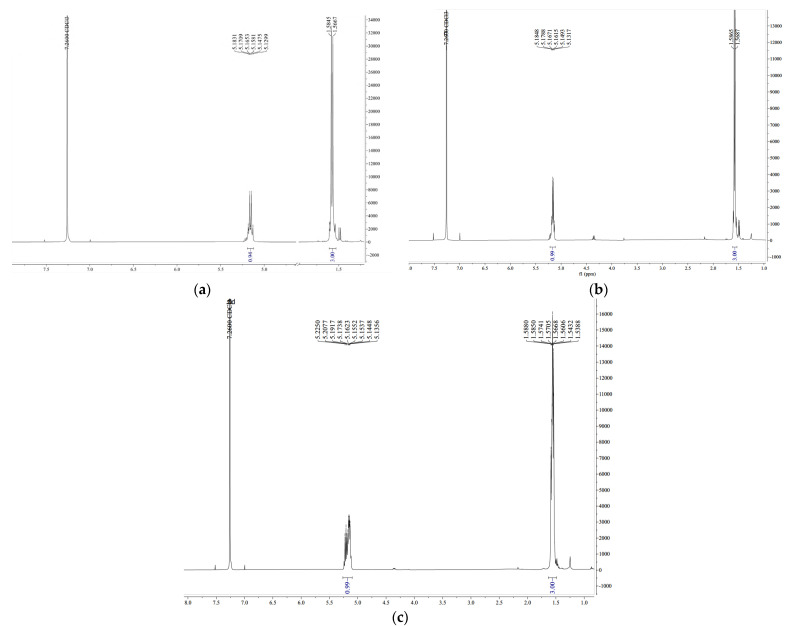
1H-RMN spectra of (**a**) purified PLA 1, (**b**) impurified PLA 2, and (**c**) lactide CPLA.

**Figure 8 ijms-24-09817-f008:**
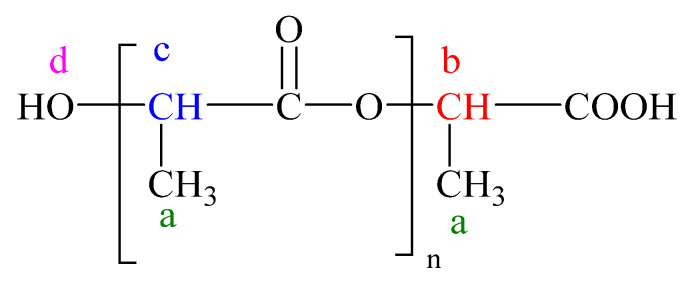
Chemical structure of the obtained PLA used for the determination of molecular weight.

**Figure 9 ijms-24-09817-f009:**
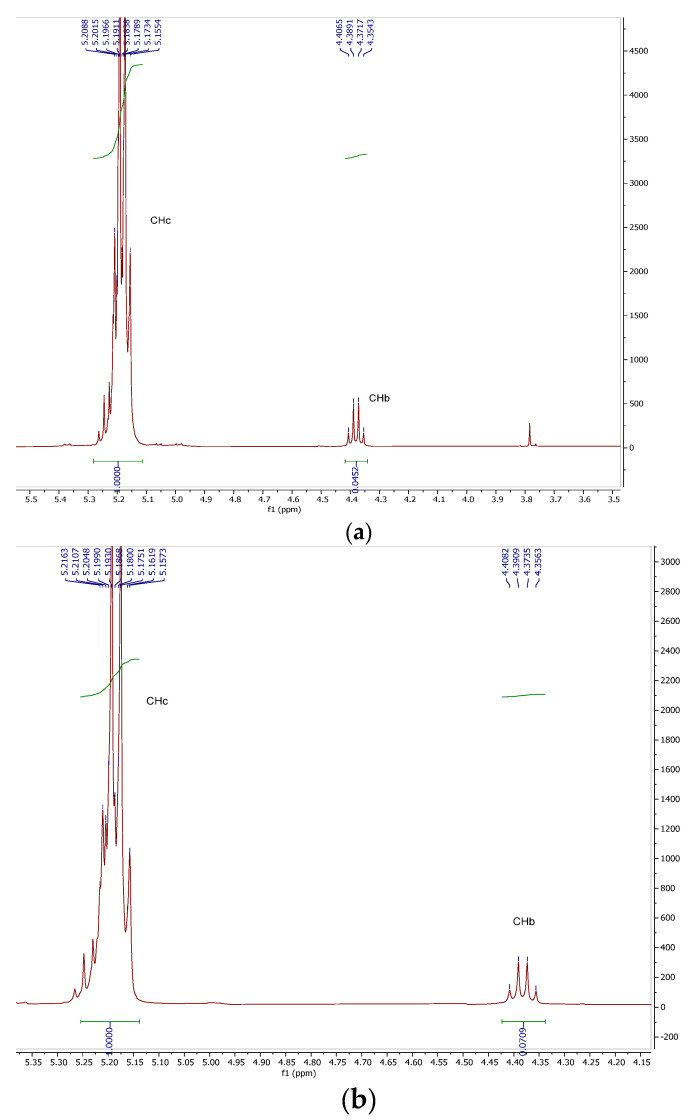
^1^H-RMN spectra of (**a**) PLA 1 and (**b**) PLA 2.

**Figure 10 ijms-24-09817-f010:**
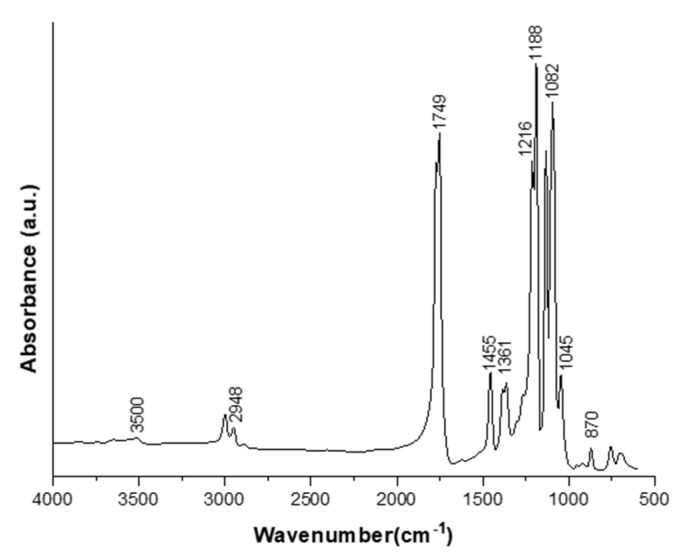
FTIR spectrum of purified PLA 1.

**Figure 11 ijms-24-09817-f011:**
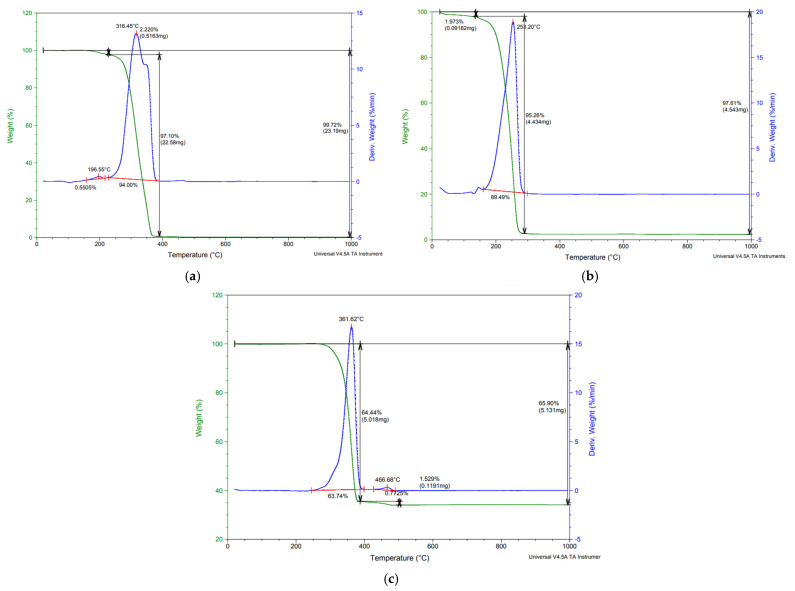
TGA/DTG curves of PLA: (**a**) impurified PLA 2, (**b**) lactide CPLA, and (**c**) purified PLA 1.

**Figure 12 ijms-24-09817-f012:**
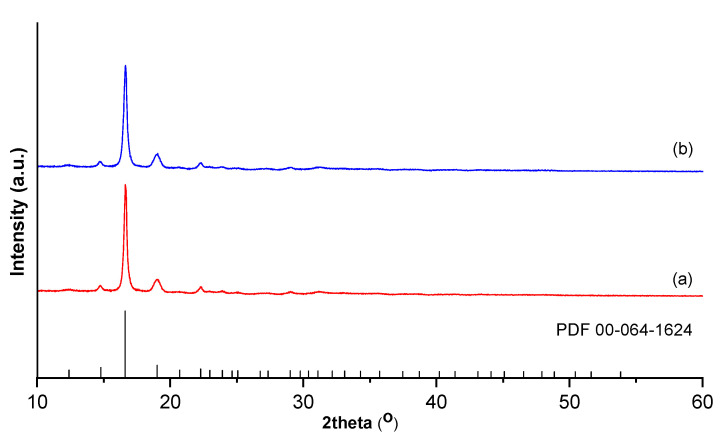
XRD patterns of (**a**) purified PLA 1 and (**b**) impurified PLA 2.

**Figure 13 ijms-24-09817-f013:**
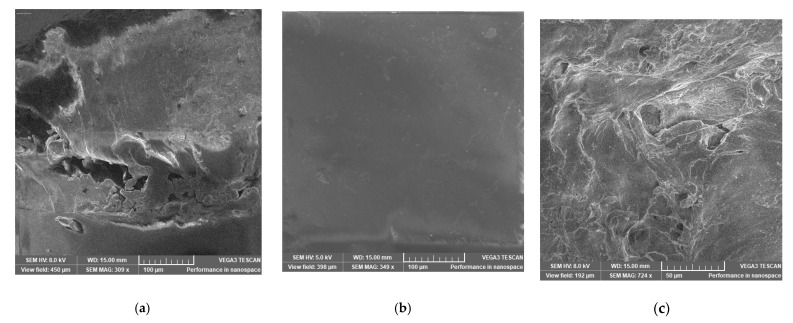
SEM micrograph analysis of PLA samples: (**a**) PLA 1, (**b**) PLA 2, and (**c**) CPLA.

**Table 1 ijms-24-09817-t001:** Content of raw, pretreated, and delignified biomass.

Raw Material	Content	Pretreated Biomass	Content	Delignified Biomass	Content
Cellulose *	38.4 ± 1.2	Cellulose **	48.5 ± 1.6	Cellulose ***	98.0 ± 1.6
Hemicelluloses *	26.8 ± 1.0	Hemicelluloses **	8.1 ± 0.04	Hemicelluloses ***	-
Lignin *	28.6 ± 0.98	Lignin **	35.8 ± 1.5	Lignin ***	2.0 ± 0.1
Solid compositions *	93.9 ± 2.2	Solid yield **	62.2 ± 1.2	Solid yield ***	± 2.4

* (g/100 g raw material, d.w.); ** (g/100 g pretreated biomass, d.w.); *** (g/100 g pretreated and delignified biomass, d.w.).

**Table 2 ijms-24-09817-t002:** Lactic acid concentration obtained in the current study and the results reported in the literature.

Raw Material	Pretreatment Used	Microorganism	L-Lactic Acid			Reference
			C (g/L)	Y (g/g)	P (g/L/h)	
Forest and marginal lands lignocellulosic	Autohydrolysis—226 °C, severity 4.15	*L. rhamnosus*	61.74	0.97	1.4	[29]
Wheat straw	Sulfuric acid pretreatment	*P. acidilactici* ZY271	107.5	0.29	2.69	[30]
Rice straw	Sulfuric acid pretreatment 1%	*Enterococcus faecium* QU 50	23.7	0.254	1.85	[31]
Rice straw	Dilute ethylenediamine	*Weizmannia coagulans* (formerly *Bacillus coagulans)*	92.5	0.58	2.01	[27]
Plum orchard waste	Pressurized hot water pretreatment	*L. rhamnosus* ATCC 7469	49.1 ± 1.1	0.93 ± 0.04	2.04 ± 0.18	This study

C—concentration of lactic acid, Y—yield, P—productivity.

## Data Availability

Not applicable.

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
