# Peer review of "L-Poly(lactic acid) Production by Microwave Irradiation of Lactic Acid Obtained from Lignocellulosic Wastes"

_ijms, 2023, doi:10.3390/ijms24129817_

Round 1
Reviewer 1 Report
This study used microwave irradiation of L-lactic acid obtained from SSF of plum orchard pruning biomass to produce L-polylactic acid (PLA), which can replace synthetic polymer in the packaging industry. Lactobacillus rhamnosus ATCC 7469 was used for L-lactic acid production, and enzymatic hydrolysis of pretreated biomass into glucose was performed using cellulase of Trichoderma reesei sp. and β-gluco-sidase from almonds. Various tests confirmed the presence of PLA, and the production process had a high productivity of 2.04 ± 0.18 g/L/h. The authors need to address the following concerns before the article is considered to be published.
1. Line 60: Please add references to “Polylactic acid (PLA)…with a global production capacity of 13.9%”.
2. The authors could add the GPC analysis if possible.
3. Figure 10: Please add the reference pattern of crystallized PLA (PDF 00-064-1624) mentioned in the text.
Reviewer 2 Report
The submitted paper entitled "L-Poly(lactic acid) production by microwave irradiation of lactic acid obtained from lignocellulosic wastes" focused on the PLA synthesis method from plum orchard waste. In my opinion, the subject of the research is novel and interesting.; however, some minor changes should be made before publication. More specific comments are listed below:
1. The article did not indicate what type of orchard waste was used, whether it was stems, leaves, or any other biomass type. Please provide some information.
2. Research focuses on confirming the PLA structure obtained in the method; however, there is no information on the properties of PLA itself. Please consider including information on the molecular weight or a comparative analysis of the rheological properties of the obtained PLA grades.
3. The authors present the SEM structure of the sample surface, but what was the general appearance of this material?
